# Traumatic Childbirth Experience and Childbirth-Related Post-Traumatic Stress Disorder (PTSD): A Contemporary Overview

**DOI:** 10.3390/ijerph20042775

**Published:** 2023-02-04

**Authors:** Leonieke Kranenburg, Mijke Lambregtse-van den Berg, Claire Stramrood

**Affiliations:** 1Department of Psychiatry, Section Medical Psychology, Erasmus University Medical Center, P.O. Box 2040, 3000 CA Rotterdam, The Netherlands; 2Department of Psychiatry and Child & Adolescent Psychiatry, Erasmus University Medical Center, P.O. Box 2040, 3000 CA Rotterdam, The Netherlands; 3Beval Beter, P.O. Box 345, 1000 AH Amsterdam, The Netherlands

**Keywords:** PTSD, childbirth, overview

## Abstract

With this manuscript we provide an overview of the prevalence, symptoms, risk factors, screening, support, and treatment for women with a traumatic childbirth experience or childbirth-related PTSD. This overview is based on both recent literature and the authors’ clinical experiences from the fields of obstetrics, psychiatry and medical psychology to provide up-to-date knowledge about recognizing, preventing and treating CB-PTSD from a clinical perspective. We pay substantial attention to prevention as there are many things health care professionals can do or not do to contribute to a positive childbirth experience, and save women, their infants and families from a sub-optimal start due to childbirth-related trauma.

## 1. Introduction

Globally, there is increasing attention to positive and traumatic childbirth experiences. The first studies were conducted in the late 1990s, and recently the World Health Organization (WHO), among others, has again highlighted the importance of a good childbirth experience [1,2,3]. Negative experiences may lead to persistent mental health issues, in particular post-traumatic stress disorder (PTSD). At the same time, pre-existing PTSD may also affect pregnancy and childbirth. Attention to both pre-existing PTSD and childbirth-related PTSD is important primarily for the mental health of the mother. Moreover, PTSD in the mother may also lead to tensions in the partner relationship [4,5] and negatively affect the relationship with and development of the child [6,7,8,9,10]. Without treatment, the mother’s symptoms may persist or can be triggered during a subsequent pregnancy. This should be prevented for the mother’s well-being, but also because intense stress during pregnancy may have negative effects on foetal development [11,12]. With this overview article, we aim to provide an up-to-date overview of various clinically relevant aspects of traumatic childbirth experience and childbirth-related post-traumatic stress disorder (CB-PTSD).

## 2. Methods

This overview is based on both recent literature and the authors’ clinical experiences from the fields of obstetrics, psychiatry and medical psychology to provide up-to-date knowledge about recognizing, preventing and treating CB-PTSD from a clinical perspective. Therefore, this overview is not the result of a thorough systematic review or meta-analysis, but aims to provide actionable insights for practicing healthcare professionals. By also relying on our clinical expertise, this of course comes together with a certain level of subjectivity, wherein some references are selected and others not. In selecting our references, we aimed to include outcomes for which most evidence is present, and when controversy exists about a certain topic, to highlight this discrepancy instead of choosing one or the other side. 

## 3. Results

### 3.1. Prevalence

Meta-analyses with data from different countries estimate the prevalence at 3.1–4.7% [13,14,15]. In high-risk groups, such as women with current depression and following infant complications, this prevalence is higher, up to 15.7% [14]. Co-morbidity is particularly frequent with depression [13]; about half of the women with childbirth-related PTSD also suffer from depression [16]. This concerns both women who already had depressive symptoms during pregnancy and thus an increased risk of childbirth-related PTSD, and women who (secondary to PTSD) developed depressive symptoms postpartum. It should be noted that childbirth-related PTSD may also occur in partners, for whom a meta-analysis showed prevalence rates of 1.2% [15].

### 3.2. Symptoms

#### 3.2.1. Traumatic Childbirth Experience

One speaks of ‘a traumatic childbirth experience’ when the woman who gave birth indicates this as such, implying that her (subjective) experience is leading. For the sake of uniformity in practice, research and education, an international consensus definition was published in 2022 [17]. This indicated that a traumatic childbirth experience (a) involves interactions between persons and/or events, (b) is directly related to childbirth, (c) caused overwhelming, stressful emotions and reactions, and (d) had short- and/or long-term impact on the woman’s health and well-being. Data from the Netherlands consistently show that about 10% of women who give birth have a traumatic experience [18,19].

#### 3.2.2. Childbirth-Related Post-Traumatic Stress Disorder (PTSD)

The DSM-5, the manual of psychiatric disorders, sets out a number of criteria for a diagnosis of PTSD [20]. The A criterion states that a person, directly or indirectly, has been exposed to actual or imminent death, serious injury or sexual violence. Next, PTSD involves the presence of four symptom clusters (B to E). These criteria are listed in Table 1, with examples applied to the peripartum period. Furthermore, for the diagnosis of PTSD, the duration of the symptoms exceeds one month, the condition causes clinically significant distress or impairment in social or occupational functioning, and the symptoms cannot be better attributed to another psychiatric disorder.

#### 3.2.3. What Makes Birth-Related Trauma Unique?

There are a number of reasons that make childbirth-related trauma different or unique compared with trauma or PTSD after experiencing other events (e.g., war, rape, accident), as listed in the paper by Horesh et al. (2021) [21]: First, having a child is socially and culturally regarded as something positive. Some therefore believe that a very intense childbirth experience should be seen more as a ‘stressful life event’, rather than a trauma. This is partly explained by the fact that in many cultures, the unpredictable course, pain, and uncertain outcome of childbirth is seen as ‘just how childbirth is’. Second, in this population, symptoms of re-experiencing are often dominant. For the birthing woman, childbirth is an event with lots of sensory inputs (sounds, smells, feelings) that can become triggers for re-experiencing the traumatic event. Third, childbirth is pre-eminently an event involving the whole family (mother, partner, baby), and therefore, also affects the whole family and inter-relationships, mother–child bonding and attachment.

### 3.3. Risk Factors

Several risk factors may lead to the development of childbirth-related PTSD [13]. These risk factors can be divided by chronology (i.e., factors that occur before, during or after childbirth) and by the nature of the risk factor (i.e., obstetric, psychological and social). Table 2 summarises these risk factors. It is also important to note that parity and place of delivery (home or hospital) are not independent risk factors for developing PTSD. The increased risk of PTSD (symptoms) in primiparous women and after hospital births may be explained by the fact that interventions are more common in these groups [19].

### 3.4. Screening and Diagnostics

Screening for predisposing risk factors, such as pre-existing PTSD due to sexual trauma or depression during pregnancy, only makes sense if the screening in itself has a positive effect and/or leads to timely available care with proven effectiveness [22]. In addition, it is advised to screen for fear of childbirth, as evidence-based treatments exist, and they are likely to have a positive effect on fear, course of childbirth, and likelihood of developing PTSD postpartum. 

When it comes to screening for birth-related trauma and PTSD, a relatively simple question to ask as a professional is “How did you experience the delivery?” In case of a negative experience or increased risk of childbirth-related PTSD, further questions can be asked. This also applies when one is not reassured by the woman’s answer to the initial question. Follow up questions may include “Would you describe your childbirth as traumatic or very upsetting?”, or “Are you currently experiencing psychological symptoms that you think are caused by childbirth?” [23]. Recognising early responses to a traumatic birth and providing advice and support can reduce the risk of PTSD developing [24]. In doing so, asking about the experience of the pre-conceptional period and the pregnancy itself is also useful. For example, if a pregnancy was preceded by one or more miscarriages, or assisted reproductive techniques (e.g., IVF), this may be related to PTSD symptoms after childbirth. The same applies to stressful events (e.g., unclear results of prenatal tests, ultrasound abnormalities) or medical episodes (e.g., recurrent vaginal blood loss) during pregnancy. 

Suitable and validated screening questionnaires include the PTSD Checklist for DSM-5 (PCL-5) [25]. Administering a questionnaire is not equivalent to making a diagnosis. The latter should be done by a mental health professional who is qualified and competent to do so, with or without the use of a validated clinical interview such as the Clinician-Administered PTSD Scale (CAPS) [26]. Although these are validated instruments for screening and diagnosis, in practice, the woman’s suffering and the clinical view of the healthcare provider is leading in whether or not to commence treatment.

### 3.5. Support

In aftercare, and also during subsequent pregnancies, there are several times when healthcare professionals (may) have a role in counselling and supporting women with childbirth-related PTSD (symptoms) or traumatic childbirth experiences.

#### 3.5.1. Obstetric Care Providers (Midwives, Obstetricians, (Labour and Maternity) Nurses)

Women may develop symptoms similar to those of PTSD in the first few days after childbirth. Often, however, these complaints are transient. In this situation, it is important to give a good explanation (psycho-education), especially that these complaints are a normal reaction to a special situation, as childbirth is not an everyday event. It is also important to monitor the complaints, so that a quick referral for treatment can be made in case symptoms persist and PTSD is suspected. 

The Netherlands is the only European country with a formal national guideline on childbirth-related PTSD [27]. Practical advice from this guideline [28] for obstetric care providers includes, in chronological order: asking the woman about her the experience of childbirth (preferably by a healthcare provider who attended or managed the delivery); then, in early postpartum days, reviewing the actual course of labour, if the woman desires, without asking extensively about negative emotions and thoughts or applying formal debriefing methods; and finally, scheduling a follow-up appointment (4–6 weeks after delivery), preferably with the person who managed the delivery. During this appointment, it is important to again discuss the delivery experience, as well as policies or wishes for any subsequent pregnancy and delivery.

#### 3.5.2. General Practitioners

General practitioners (family physicians) may see women at their practice who, shortly or longer after childbirth, report symptoms of PTSD, depression or an anxiety disorder. Distinguishing between these is crucial, at least because the treatment differs. In any woman presenting with symptoms of (postpartum) depression, it is important to ask about the course and (more importantly) the experience of childbirth; just as it is relevant to ask about symptoms of depression and suicidality in the case of childbirth-related trauma or PTSD. Distinguishing between physiological consequences of life-events, and psychological symptoms/disorders is a sometimes difficult, but an essential task of the professional with whom women discuss this.

#### 3.5.3. Paediatricians and Infant Health Services

Doctors and nurses caring for new-borns and children not only have a crucial role in monitoring a child’s development, but can also contribute to identifying psychological symptoms in parents. Asking about the experience of childbirth (and not just the medical course) is recommended. Especially with babies who have difficulty sleeping, with feeding problems, excessive crying, and restless or anxious behaviour [8], asking about their mothers’ delivery experience, psychological complaints and (psychosocial) stress factors is essential. Referral to an infant mental health specialist can also be very useful in such cases to safeguard a healthy mother–baby relationship in stressful circumstances.

### 3.6. Treatment

Evidence-based treatments for PTSD, as recommended in various guidelines, are trauma-focused cognitive behavioural therapy (CBT) and eye movement desensitisation and reprocessing (EMDR) [29,30,31]. A key component of trauma-focused cognitive behavioural therapy is exposure [32]. This involves both imaginary exposure—recalling the memory of the traumatic event with eyes closed as if one were back in the event—and exposure in vivo, such as revisiting the delivery rooms. EMDR treatment also involves recalling the memory of the traumatic event. The patient is asked to focus on the memory image that evokes the most tension at present, while simultaneously performing a task that taxes working memory (such as following moving lights on a light bar with the eyes). This creates the opportunity to process the traumatic event, while the emotional load subsides and renewed meaning-giving becomes possible [33]. EMDR has the advantage that no homework assignments need to be completed between sessions, which is an upside for women who just gave birth and have little time to themselves anyway [34]. With EMDR, symptom reduction can sometimes occur after just one treatment session [23]. However, treating women with a history of abuse or neglect, which was triggered or worsened by the delivery, will often require more time and/or additional forms of therapy. 

It was long thought that trauma-focused treatment during pregnancy would be too stressful or even dangerous. However, recent research shows no evidence for this [35,36]. The rationale for the use of trauma-focused psychotherapy in pregnant women with (childbirth-related) PTSD, is that a trade-off needs to be made between the tension evoked by the treatment itself and the continuous tension/stress resulting from the PTSD during the rest of the pregnancy—and its impact on the delivery and foetus—if the symptoms were not treated. 

All in all, trauma-focused psychological treatments for PTSD have been found to be more effective than drug treatment [37]. However, there are reasons to consider drug treatment as a first step nonetheless, particularly if there is no possibility of starting trauma-focused psychological treatment yet, and there is exhaustion due to severe lack of sleep (related to PTSD) or severe hyperarousal symptoms with impulse breakthroughs. Furthermore, in cases of severe co-morbid depression including suicidality, the advice is to start treatment with antidepressants [37].

### 3.7. Prevention

With regard to prevention, a distinction can be made between primary prevention (=of the traumatic childbirth experience) during pregnancy and childbirth, and secondary prevention (=of PTSD (symptoms) after a traumatic childbirth experience) during puerperium. Advice for each type of prevention is listed below.

#### 3.7.1. Primary Prevention

During pregnancy, important issues include:-Identification of at-risk women. This includes women with pre-existing psychopathology and previous traumatic experiences. Study results show that identifying these women had an impact on how healthcare professionals interacted with them, and that symptoms of re-experiencing were less common than before introduction of this method in this group [22];-Education and preparation. Realistic expectations of the course of childbirth, the likelihood of common procedures, interventions and complications occurring, and attention to the good luck/bad luck factor (as opposed to success/failure) can contribute to how women experience childbirth;-Birth plan. Formulating wishes around childbirth helps women, partners and healthcare providers identify what issues they consider important. In contrast to the ‘factual’ questions/issues formulated in many birth plans, we recommend emphasising wishes related to communication, decision making, recognising stress or fear and supportive measures in this respect.

During childbirth, important issues include:-Attention to communication, explanation, informed consent, emotional support and that the woman feels seen and heard. In Dutch research among more than 2000 women with a traumatic childbirth experience, they indicated these issues as the most important things healthcare providers could have done to prevent their traumatic experience [38];-Trauma-informed care. This includes the notion that caregivers are aware that having experienced one or more traumatic events earlier in life, will have an effect on their patient’s thinking, feeling and doing [39,40]. Related to this, it is important that the woman discusses with her caregiver what contributes to her feeling safe and respected;-Continuity of care and of caregiver. There is an abundance of research showing that women who receive continuous care from a known and trusted provider are more likely to have positive obstetric outcomes and less likely to have a negative birth experience [41]. In particular, women with pre-existing trauma and with many medical interventions during childbirth are less likely to have a traumatic birth experience if they feel supported [42];

#### 3.7.2. Secondary Prevention

Secondary prevention is partly about issues already discussed in the paragraph above on support, but also about enhanced alertness and early action in case of factors that increase the risk of developing childbirth-related PTSD, such as feeling that one’s own life or that of the baby has been at risk, lack of social support, and poor coping skills [43]. 

There is increasing research on the possible advantages and disadvantages of (very) early intervention. On the one hand, one does not await the (commonly) favourable natural course of acute stress symptoms, and medicalisation/overtreatment is likely. On the other hand, many women indicate a need for and benefit from treatment of their symptoms shortly after childbirth. A recent review and meta-analysis concluded that early (<3 days postpartum) psychological interventions reduced the (severity of) PTSD symptoms 4 to 6 weeks after childbirth [44].

## 4. Conclusions

With this manuscript we have provided an overview of the prevalence, symptoms, risk factors, screening, support, and treatment for women with a traumatic childbirth experience or childbirth-related PTSD. In the previous section, we have paid substantial attention to prevention, as there are many things health care professionals can do or not do to contribute to a positive childbirth experience, and save women, their infants and families from a sub-optimal start due to childbirth-related trauma.

## Figures and Tables

**Table 1 ijerph-20-02775-t001:** PTSD criteria according to DSM-5, with examples from the pregnancy and childbirth domain.

TRAUMA
**A. One is, directly or indirectly, subjected to actual or threatened death, serious injury and/or sexual violence.**
Pregnancy-specific examples include serious concerns about the well-being of the baby and/or mother: foetal distress, pathological decelerations of foetal heart rate, umbilical cord prolapse, preterm birth, uterine scar rupture, shoulder dystocia, placental abruption, emergency caesarian section, vacuum/forceps deliveries and postpartum haemorrhage.
Note: One of the things that complicates making a PTSD diagnosis after childbirth is the subjective nature of the A criterion. For example, the medical professionals’ assessment that the woman in question or her baby are not in mortal danger may not necessarily correspond to her own subjective experience of “I thought I was dying” or “I feared the death of my baby”. It may also be that the childbirth experience somehow touches on the memory of previous traumatic experiences in a person’s life, such as sexual abuse, leading to a recurrence of symptoms. Exposure and touching of the genital area by professionals, and experiencing pain and loss of control, inevitably linked to childbirth, may then act as a trigger of previous sexual trauma.
**SYMPTOMS**
**B. Re-experiencing**
Pregnancy-specific examples: nightmares about childbirth, or feeling that one keeps reliving (part of) the birth experience, physical reactions when watching or reading something about pregnancy or birth, or when visiting the hospital for postpartum check-up. Pre-existent PTSD examples: re-living unpleasant or unwanted touch or penetration, being restrained or unable to move.
**C. Avoidance**
Pregnancy-specific examples: attempts to avoid (answering) the question “how was your delivery?” as much as possible, or by avoiding things reminiscent of birth, such as visiting the hospital. In extreme cases, this may also lead to care avoidance and missing out on necessary care. After a traumatic delivery, the genitals may also be a constant reminder of the trauma, causing the mother to avoid intimate contact with her partner or gynaecologic examinations.
**D. Negative changes in mood and cognition**
Pregnancy-specific examples: feeling excessively guilty after a caesarean section, augmentation of labor or choosing pain relief because “I didn’t manage to give birth ‘normally/without help’”, and thinking “I failed”; or a persistent negative state of mind, with feelings of fear, disgust, anger, guilt or shame being in the foreground since childbirth.
**E. Hyperarousal**
Pregnancy-specific examples: irritable behaviour, often towards the partner; or excessive alertness so that nothing happens to the baby, such as constantly checking whether the baby in the cradle is still breathing.

**Table 2 ijerph-20-02775-t002:** Risk factors for childbirth-related PTSD.

	Antepartum	Peripartum	Postpartum
Obstetric	- Complicated course of pregnancy, e.g., preterm birth, (pre-)eclampsia, HELLP syndrome	- Complicated course of delivery, e.g., emergency caesarean section, ventouse delivery, postpartum haemorrhage	- Complications with the neonate, e.g., NICU admission, resuscitation, perinatal mortality
Psychological	- Psychiatric history - Previous psychotrauma (e.g., sexual violence) - Fear of childbirth (tocophobia) - Depression during pregnancy	- Experience of delivery, e.g., of the treatment and communication by professionals - Dissociation	- Postpartum depression - Stress and poor coping skills
Social	- Psychosocial vulnerability, e.g., (initial) unwanted pregnancy, stressful circumstances (relationship, finances, housing, work, health)	- Lack of practical and emotional support	- Lack of practical and emotional support, e.g., lack of understanding/no recognition of severity of symptoms

## Data Availability

Not applicable.

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
