# Peer review of "Traumatic Childbirth Experience and Childbirth-Related Post-Traumatic Stress Disorder (PTSD): A Contemporary Overview"

_ijerph, 2023, doi:10.3390/ijerph20042775_

Round 1
Reviewer 1 Report
Thank you for the opportunity to review this contemporary overview of childbirth related PTSD. The article is generally well-written, and I very much appreciate that the authors understand the literature related to not only childbirth itself as potentially traumatic exposure capable of eliciting a trauma response such as PTSD but also that childbirth experiences can trigger existing PTSD symptomatology – which is a very important point that has been overlooked in many instances in the traumatic CB literature.
As a perinatal clinician and scholar, I agree with the general sentiments reflected in this paper. However, it is unclear as written how several of the assertions are supported in the literature or whether many of these are in fact the author’s own views based on clinical experiences & expertise. Is this indeed a review article? Or is it more of a commentary? Because if this is a review article, more detail is needed regarding the methodology employed for conducing this review, and some reflexivity is needed to better understand the positionality of the authors for those passages that are based solely on the authors’ clinical expertise and not otherwise supported in the literature. Specific questions I have are:
What methods were used for collecting the included reviewed literature? Time frame?
Inclusion/exclusion criteria? How many articles were reviewed and of what type?
Was a specific review framework applied (narrative review, scoping review, etc.)?
How many investigators conducted the review? Were there discrepancies between authors regarding which articles to include in the review? If so, how were these resolved? The addition of a reflexive note about the authors would be helpful for understanding any specific influences undergirding the design and conduct of the review, and for understanding when and how the article is reliant solely on the clinical experiences and expertise of the authors.
What screening and analytic process(es) were applied?
A variety of types of articles were included. Were any systematic appraisals of methodological quality or confidence in the findings of the included studies undertaken (e.g., JARS, Critical Appraisal Skills Programme, GRADE, etc)?
Understanding whether this is a review article or a commentary based on the authors’ clinical expertise is important for understanding the voracity of their claims. One example of how this makes it hard to interpret the content of the article is in the section on screening & diagnostics. Regarding whether or not to screen, the authors cite one article from 2014; the literature on this is more complicated and as such seems incomplete if this is indeed a review article. If it is not a review article but rather a commentary, there is more latitude – however it needs to be clearly identified as such.
Additional observations/queries:
Page 1, Line 35 the “A” following the semi-colon should not be capitalized.
Table 1 is very useful clinically due to the addition of perinatal-specific examples. How were these examples derived? Are they grounded in the literature? Should emergency c-section, vacuum/forceps deliveries be added as examples?
There seem to be many instances where extra spaces are in between words and sentences that should be eliminated.
Author Response
Dear editor and reviewers,
Thank you for your constructive comments. We believe these are helpful in improving our paper. In the text below we have addressed these comments point-by-point, followed by the changes that we have made to the manuscript.
Reviewer 1
*Thank you for the opportunity to review this contemporary overview of childbirth related PTSD. The article is generally well-written, and I very much appreciate that the authors understand the literature related to not only childbirth itself as potentially traumatic exposure capable of eliciting a trauma response such as PTSD but also that childbirth experiences can trigger existing PTSD symptomatology – which is a very important point that has been overlooked in many instances in the traumatic CB literature.
Reply: Thank you for these compliments.
*As a perinatal clinician and scholar, I agree with the general sentiments reflected in this paper. However, it is unclear as written how several of the assertions are supported in the literature or whether many of these are in fact the author’s own views based on clinical experiences & expertise. Is this indeed a review article? Or is it more of a commentary? Because if this is a review article, more detail is needed regarding the methodology employed for conducing this review, and some reflexivity is needed to better understand the positionality of the authors for those passages that are based solely on the authors’ clinical expertise and not otherwise supported in the literature.
Reply: We do understand these questions of the reviewer. In short, assertions made in the paper are supported by literature. In doing so, we aimed to select the most recent articles and meta-analyses and make references to these articles. As all authors are experienced in the field of peripartum mental health care and have backgrounds in various disciplines (i.e. Gynecology-Obstetrics, Psychiatry, Medical Psychology) both from a clinical and a research perspective, we have a profound overview of the literature. For this paper, we have aimed to make a relevant and up-to-date selection of the literature. By also weighing in our clinical expertise, this of course comes together with a certain aspect of arbitrariness, wherein some references are made and others not. In selecting our references, we aimed to include outcomes for which most evidence is present, and, in case controversy exists about a certain topic, to name this discrepancy instead of choosing one or the other side. In addition to the above, it is true that the paper does reflect our clinical experience as well. We believe this is also necessary and desirable, given our aim to provide actionable insights for practicing healthcare professionals. Combining literature and clinical experience, one could say that our paper is more of an overview or update or clinical review, and not a (systematic) review article in a strict methodological sense. This is also why we chose the wording “overview” in our title, and do not mention the word “review” anywhere in our paper. However, as explained above, we do understand the reviewer’s point and in order to avoid confusion, we have now added a clarifying text to the paper that explains the purport and scope of the paper.
Changes made to the manuscript:
Added to introduction: “With this overview article we aim to provide an up-to-date overview of various clinically relevant aspects of traumatic childbirth experience and childbirth-related post-traumatic stress disorder (CB-PTSD). This overview is not the result of a thorough systematic review or meta-analysis, but is based on both recent literature and the authors’ clinical experiences from the fields of obstetrics, psychiatry and medical psychology to provide up-to-date knowledge about recognizing, preventing and treating CB-PTSD from a clinical perspective .”
*Specific questions I have are:
What methods were used for collecting the included reviewed literature? Time frame?
Inclusion/exclusion criteria? How many articles were reviewed and of what type?
Was a specific review framework applied (narrative review, scoping review, etc.)?How many investigators conducted the review? Were there discrepancies between authors regarding which articles to include in the review? If so, how were these resolved? The addition of a reflexive note about the authors would be helpful for understanding any specific influences undergirding the design and conduct of the review, and for understanding when and how the article is reliant solely on the clinical experiences and expertise of the authors.
What screening and analytic process(es) were applied?
A variety of types of articles were included. Were any systematic appraisals of methodological quality or confidence in the findings of the included studies undertaken (e.g., JARS, Critical Appraisal Skills Programme, GRADE, etc)?
Reply: These are valid questions, if our paper were to be intended as a systematic review. As we explain above, this was not the case. As the scope of our paper is quite broad and clinically oriented, we also weighed in the clinical applicability of data and recommendations. For some paragraphs (e.g. prevalence, risk factors) more solid and quantitative evidence is available than for other topics (e.g. support, prevention) for which we have relied more on clinical expertise,
*Understanding whether this is a review article or a commentary based on the authors’ clinical expertise is important for understanding the voracity of their claims. One example of how this makes it hard to interpret the content of the article is in the section on screening & diagnostics. Regarding whether or not to screen, the authors cite one article from 2014; the literature on this is more complicated and as such seems incomplete if this is indeed a review article. If it is not a review article but rather a commentary, there is more latitude – however it needs to be clearly identified as such.
Reply: Thank you again for pointing this out, as it is an important point. To avoid confusion about the methodology of our paper, we have adjusted the text. Furthermore, we updated the references regarding screening, by including the paper from Slade e al, published in the BMJ In 2022.
Changes made to the manuscript:
Added to introduction: “With this overview article we aim to provide an up-to-date overview of various clinically relevant aspects of traumatic childbirth experience and childbirth-related post-traumatic stress disorder (CB-PTSD). This overview is not the result of a thorough systematic review or meta-analysis, but is based on both recent literature and the authors’ clinical experiences from the fields of obstetrics, psychiatry and medical psychology to provide up-to-date knowledge about recognizing, preventing and treating CB-PTSD from a clinical perspective.”
Added to the paragraph screening & diagnostics: “Recognising early responses to a traumatic birth and providing advice and support can reduce the risk of PTSD developing”, and make reference to Slade P, Murphy A, Hayden E. Identifying post-traumatic stress disorder after childbirth. BMJ. 2022 May 10;377:e067659. doi: 10.1136/bmj-2021-067659.
*Additional observations/queries:
Page 1, Line 35 the “A” following the semi-colon should not be capitalized.
Reply: we have adjusted this.
Changes made to the manuscript:
We replaced the A with a.
*Table 1 is very useful clinically due to the addition of perinatal-specific examples. How were these examples derived? Are they grounded in the literature? Should emergency c-section, vacuum/forceps deliveries be added as examples?
Reply: Thank you for the compliment about usefulness. These examples were derived from clinical practice. We agree with the question to include emergency c-section, vacuum/forceps deliveries as examples as well, and have done so in the new version of the manuscript.
Changes made to the manuscript:
Table 1: we have added emergency c-section, vacuum/forceps deliveries as examples to Table 1.
There seem to be many instances where extra spaces are in between words and sentences that should be eliminated.
Reply: Thank you for notifying, we have corrected this.
Changes made to the manuscript:
We have deleted extra spaces throughout the manuscript.
Reviewer 2 Report
Novelty and methodology are not clear. Authors should explain these crucial items in the publication.
Author Response
Dear reviewer,
Thank you for your constructive comments. We believe these are helpful in improving our paper. In the text below we have addressed you comments, followed by the changes that we have made to the manuscript.
Reviewer 2
*Novelty and methodology are not clear. Authors should explain these crucial items in the publication.
Reply: Thank you for addressing this point, as this is important information for the reader in interpreting this manuscript. The manuscript is an overview article and not a systematic review. As this area of research has under continuous development and progress, we perceived this a suited point in time for a contemporary overview from a clinical perspective. As such, the novelty lies in including up-to-date literature, and in providing a comprehensive overview of the topic ranging from prevalence to risk factors, from prevention to treatment etc.. As all authors are experienced in the field of peripartum mental health care and have backgrounds in various disciplines (i.e. Obstetrics-Gynecology, Psychiatry, Medical Psychology) both from a clinical and a research perspective, we have a profound overview of the literature. For this paper, we have aimed to make a relevant and up-to-date selection of the literature. By also weighing in our clinical expertise, this of course comes together with a certain aspect of arbitrariness, wherein some references are made and others not. In selecting our references, we aimed to include outcomes for which most evidence is present, and, in case controversy exists about a certain topic, to name this discrepancy instead of choosing one or the other side. In addition to the above, it is true that the paper does reflect our clinical experience as well. Combining literature and clinical experience, one could say that our paper is more of an overview or commentary and not a review article in a strict methodological sense. This is also why we chose the wording “overview” in our title, and do not mention the word “review” anywhere in our paper. Furthermore, and as reviewer 3 notices, our aim is to provide actionable insights provided for a wider range of health care providers However, as explained above, we do understand the reviewer’s point and in order to avoid confusion, we have now added a clarifying text to the paper that explains the purport and scope of the paper.
Changes made to the manuscript:
Added to introduction: “With this overview article we aim to provide an up-to-date overview of various clinically relevant aspects of traumatic childbirth experience and childbirth-related post-traumatic stress disorder (CB-PTSD). This overview is not the result of a thorough systematic review or meta-analysis, but is based on both recent literature and the authors’ clinical experiences from the fields of obstetrics, psychiatry and medical psychology to provide up-to-date knowledge about recognizing and treating CB-PTSD from a clinical perspective." .”
Reviewer 3 Report
Interesting article about PTSD & traumatic childbirth experience. Actionable insights provided for general practicioners as well as other health care providers
Author Response
Dear editor and reviewers,
Thank you for your constructive comments. We believe these are helpful in improving our paper
Reviewer 3
*Interesting article about PTSD & traumatic childbirth experience. Actionable insights provided for general practicioners as well as other health care providers
Reply: Thank you for your positive comments.
Round 2
Reviewer 1 Report
I have reviewed the revisions and do believe the manuscript has been sufficiently improved to warrant publication in IJERPH (following final copy editing).
Author Response
Dear reviewer,
Thank you, we are happy to read that the manuscript has been sufficiently improved to warrant publication in IJERPH.